# Experiences of transgender and non-binary youth accessing gender-affirming care: A systematic review and meta-ethnography

**Seán Kearns**[1,2]*, **Thilo Kroll**[4‡], **Donal O'Shea**[1,2,3‡], **Karl Neff**[1,2]

**1** School of Medicine, University College Dublin, Dublin, Ireland, **2** National Gender Service, St Columcille's Hospital, Dublin, Ireland, **3** Department of Endocrinology and Diabetes Mellitus, St Vincent's University Hospital, Dublin, Ireland, **4** Centre for Interdisciplinary Research, Education and Innovation in Health Systems, School of Nursing, Midwifery and Health Systems, University College Dublin, Dublin, Ireland

☯ These authors contributed equally to this work.
‡ These authors also contributed equally to this work.
* sean.kearns@ucd.ie, sean.kearns1@ucdconnect.ie

## Abstract

### Objective

Transgender and non-binary individuals frequently engage with healthcare services to obtain gender-affirming care. Little data exist on the experiences of young people accessing gender care. This systematic review and meta-ethnography aimed to identify and synthesise data on youths' experiences accessing gender-affirming healthcare.

### Method

A systematic review and meta-ethnography focusing on qualitative research on the experiences of transgender and non-binary youth accessing gender care was completed between April-December 2020. The following databases were used: PsychINFO, MEDLINE, EMBASE, and CINAHL. The protocol was registered on PROSPERO, international prospective register of Systematic Reviews (CRD42020139908).

### Results

Ten studies were included in the final review. The sample included participants with diverse gender identities and included the perspective of parents/caregivers. Five dimensions (third-order constructs) were identified and contextualized into the following themes: 1.) Disclosure of gender identity. 2.) The pursuit of care. 3.) The cost of care. 4.) Complex family/caregiver dynamics. 5.) Patient-provider relationships. Each dimension details a complicated set of factors that can impact healthcare navigation and are explained through a new conceptual model titled "The Rainbow Brick Road".

### Conclusion

This synthesis expands understanding into the experience of transgender and non-binary youth accessing gender-affirming healthcare. Ryvicker's behavioural-ecological model of

**Data Availability Statement:** All relevant data are within the manuscript and its Supporting Information files.

**Funding:** The author(s) received no specific funding for this work.

**Competing interests:** The authors have declared that no competing interests exist.

healthcare navigation is discussed in relation to the findings and compared to the authors' conceptual model. This detailed analysis reveals unique insights on healthcare navigation challenges and the traits, resources, and infrastructure needed to overcome these. Importantly, this paper reveals the critical need for more research with non-binary youth and research which includes the population in the design.

## Language and definitions–"a" definition not "the" definition

As our understanding of the complexities of gender and sex continues to develop, so too has our understanding of language and terminology. See Table 1 for a brief overview of language usage past and present.

## Introduction

For the purpose of this paper, the terms transgender, trans and non-binary will be used to encompass the spectrum of gender identities that represent participants across the studies. The authors acknowledge the ever-changing nature of language and gender identity, and recognize that one definition cannot fit all adequately. For this review, the authors will be exploring the experiences of transgender and non-binary young people accessing gender-affirming care as a gender non-conforming young person experiencing gender dysphoria.

Researchers have struggled to produce accurate statistics on the prevalence of transgender people [12, 13]. In the past, the DSM (Diagnostic and Statistical Manual) has categorised "transgender identification" as "apparently rare" and "apparently uncommon" [13]. In 2015, a systematic review of 12 studies reported a prevalence of 4.6 per 100,000 or 1 in every 21,739 individuals [14].

However, since then it has become very clear that the incidence of gender incongruence is rising [15]. This is largely due to the increase in the number of adolescents seeking gender affirming interventions over the last decade [16]. The demographics of adolescents presenting for intervention are also changing over time, with more people who were assigned female at birth presenting to services than people who were assigned male at birth [17].

Significant methodological limitations exist in the collection of prevalence data. Non-binary individuals are often underrepresented as most population health surveys only collect binary gender data, which makes it difficult to gain reliable estimates of gender affirming health care. Given the methodological limitations, it is not surprising that the prevalence of non-binary identities can vary across studies [18, 19]. Similarly, for prevalence data, if data points are collected during healthcare encounters, then transgender individuals' who do not use gender affirming hormone therapy or who have not had gender affirming surgery will not be represented in the outcomes.

Literature strongly highlights that systems frequently fail transgender individuals, even in countries with progressive civil liberties [20–42]. Multiple studies describe the arduous experiences of trans adults seeking healthcare [43–52]. Notably, most will report experiences of stigma and discrimination when accessing healthcare, from encounters with administrative staff to encounters with healthcare providers [53–58].

Stigma has been detailed as both the overt refusal of care and more subtle stigmatisation through refusal to use affirming language [54]. Erasure has been detailed in terms of institutional/structural erasure as well as informational, showcasing a lack of policies to accommodate trans identities or an overt lack of acceptance of the need for structural changes. Further barriers to gender affirming care include a reluctance to disclose identity [59], financial

**Table 1. Glossary of terms.**

| | |
|---|---|
| Transsexual | "Transsexual" coined in the 1949 essay by Dr. Cauldwell "Psychopathia Transexualis" and was used to describe a person whose gender assigned at birth, based on the appearance of their external genitalia, did not match their gender identity [1]. |
| | Transsexualism was used to describe the condition in which a person's gender identity was not the same as the gender they were assigned at birth, and historically referred mainly to binary transgender people who had sought medical assistance to affirm their gender [2]. While transsexualism is no longer typically used in specialist gender services, the term is still sometimes used in healthcare and psychology disciplines. This term can be contentious in social realms due to historical connotations with mental illness and therefore should not be assumed to be appropriate for general use. |
| Transgender Trans* | "Transgender" is the more commonly used umbrella term to describe the gender of a person whose gender identity or gender expression differs from the normative societal expectations of the gender they were assigned at birth [3]. Transgender became used colloquially among queer communities in 1971 and the shortened term "trans" or "trans*" is as contemporary as 1996 [4]. Transgender individuals may identify in a binary manner, meaning, if their assigned gender was female, they identify as male (sometimes documented as "FtM"), and vice versa ("MtF"). The person may use the term "trans man", "trans woman", "transgender man" or "transgender woman". Assigned female at birth (AFAB) and assigned male at birth (AMAB) are the current preferred terms used vs "FtM" or "MtF". |
| Non-Binary | "Non-binary" is another umbrella term that refers to individuals who do not identify as exclusively male/masculine or female/feminine [5, 6]. A non-binary individual may identify with neither of these, fluctuate between the gender binary, identify with both, or reject all. Non-binary people may identify as non-binary, genderfluid, gender flex, gender non-conforming, gender non-normative or genderqueer. More recently, there has been an increase in quantitative research of people rejecting the binary. There is inherent overlap between trans and non-binary identities with some individuals having affinity to one term alone or both (for example, a non-binary person may also identify as trans). |
| Gender nonconformity | "Gender nonconformity "is a term used when a person's gender identity, role, or expression does not align with the socially sanctioned expectations of a set gender [7]. |
| Gender dysphoria | Gender dysphoria is a condition of psychological distress due to an incongruence between a person's gender and the gender that they were assigned at birth [8]. A person may be gender nonconforming their entire life without experiencing gender dysphoria. If one does experience dysphoria, then this can be alleviated by gender affirming medical interventions, such as gender affirming hormone therapy and gender affirming surgery. Both binary transgender individuals and non-binary individuals may benefit from medical intervention [9]. As well as medical intervention, gender affirming interventions based on social transition (changing their gender expression in social situations to align with their gender identity) and psychosocial support can effectively remediate gender dysphoria [10, 11]. |

barriers [60], insurance coverage denials [61], age (as additional screening for younger ages and need for parental consent are often requirements) [62], concerns over quality [63] and geographical constraints [64]. In many instances transgender individuals have been unable to access gender-related care [65]. Moreover, many individuals have self-reported being refused general healthcare due to their gender identity.

The United States National Transgender Discrimination Survey reported that 19% of trans individuals were refused care [18]. Due to discriminatory actions of health providers, many trans individuals avoid interaction with them [45, 66] and have sourced medication and care from unregistered sources [67]. In some urban areas, it has been found 29–63% of trans Americans were sourcing hormones from disreputable sources [68]. This phenomenon is likely to be largely driven by the difficulty in navigating healthcare systems via a biomedical model that often reinforces a pathological view of gender identity. Such a system leaves many trans individuals feeling ostracised and establishes a patient-provider power dynamic often referred to as "gatekeeping" [69–71].

While there are many peer-reviewed papers on the experiences of transgender adults accessing healthcare [71–79], there is a paucity of research around young trans and non-binary

people's experiences navigating gender-specific healthcare [80]. Globally, rates of referral to services providing gender care are rising and, as outlined above, adolescents are at the forefront of the rise. It is important to understand young people's experiences and health care and support needs, and to respond to these. This may help us to break down barriers and better support trans and non-binary youth who are seeking gender affirming medical interventions.

Therefore, the aims of this study were:

1. to systematically search, retrieve, and appraise the qualitative empirical literature on the experiences of young transgender and non-binary youth accessing healthcare.

2. to construct a new line of argument/conceptual model based on this literature

3. to synthesize and discuss the results through the lens of both this new conceptual model and Ryvicker's existing model of behavioural ecological perspective.

## Method

### Design

A meta-ethnography and synthesis as detailed by Noblit and Hare was performed [81]. This is a systematic process of interpreting qualitative data by translating concepts and metaphors across studies, often with the aim of producing new theories or conceptual models. This method was chosen as it facilitates a holistic understanding of the chosen phenomenon whilst progressing knowledge generation through its advanced synthesis. Exclusively qualitative research was chosen as it was best positioned to answer the research question surrounding young people experiences of healthcare access while allowing interpretive approaches for new theory generation. The seven-stage method was employed to collate data and work towards the generation of new understanding [82]. The project was registered on Prospero | (Registration number: CRD42020139908). The information assimilated was reported via the Enhancing Transparency in Reporting the Synthesis of Qualitative Research (ENTREQ) framework [83] (S1 File).

### Search strategy

A systematic search was completed across four databases: PsycINFO, CINAHL, EMBASE, and MEDLINE. These databases were chosen as they are the major medical, nursing, allied health and psychiatry databases which are relevant to the study topic. The search terms were informed by the PEO framework (Population, Exposure, Outcomes) [84]. This framework is a useful format for de-constructing qualitative research questions and can be used to shape a project search strategy. It involves identifying the population, the set exposure, and the designated outcome. For this study the population was represented by keywords synonymous with "transgender and non-binary youth", exposure with "healthcare" keywords and outcomes represented by "experiences/perceptions". The search was completed in April 2020 with MeSH terms and Boolean operators used where appropriate (S2 File), these helped to focus the search to more relevant papers. The search strategy was informed by previous systematic reviews on transgender healthcare [76, 78, 79]and aside from restricting the search to studies published in English, no limits were placed on the search. KN and SK confirmed the search strategy prior to completion.

### Inclusion/exclusion

Studies were included if they were primary studies with a qualitative or mixed methods research design that explored the experiences of young trans or gender non-conforming persons accessing gender-affirming care. Youth was defined as per the World Health

Organisation (WHO) as 12–24 years old [85]. Studies that reported on the views of multiple stakeholders were included if they provided qualitative data regarding young people's experiences that could be extracted separately. Mixed-methods were considered if qualitative data could be extracted. Studies where the age of the sample was not clearly reported or where data were not trans specific were excluded, i.e. studies that included LGBTQ samples were not included, only studies whose entire sample were trans/non-binary.

### Screening

All study titles and abstracts were reviewed by SK. KN then independently reviewed over 20% of all the titles and abstracts (n = 297). Screening was completed using Covidence software. Discrepancies were resolved through discussion at this stage, and consensus was achieved with sufficient inter-reliability (κ = 0.82). A third reviewer DOS was available to resolve any queries if consensus could not be met, this was not needed (n = 0). Full-text articles were reviewed in full (n = 40) by SK and KN and DOS were available for further consultation as needed. An acceptable inter-reliability was reached for full-text articles (κ = 0.87) and ten studies were included after final consultation.

### Quality appraisal

The quality of studies chosen for inclusion was assessed using the Critical Appraisal Skills Programme (CASP) checklist [86]. This is a checklist specifically designed for the appraisal of qualitative research. SK and KN reviewed the articles independently for inclusion. Mixed-methods studies were assessed using the qualitative CASP checklist as only qualitative data were used in the synthesis.

### Data extraction and data synthesis

The synthesis was informed by the seven phases of meta-ethnography originally described by Noblit and Hare. The analysis aims to create third-order constructs or themes from first-order constructs (respondents' quotations) and second-order constructs (authors' interpretation).

Phase one titled "getting started" relates to assessing if the qualitative synthesis is needed, assessing if you have the right people involved, and is there a clear research aim. There is a dearth of qualitative reviews in this area and the research team involves a strong cohort of researchers from different backgrounds with expertise in this field as well as clinicians and specialist nurses who work in the area of gender healthcare.

The second phase "deciding what is relevant" involves creating a search strategy, inclusion and exclusion criteria, deciding an appraisal tool, and implementing the search strategy and quality check. This was completed by SK and KN.

Phase three involved "reading the studies" and identifying first order and second-order constructs. The studies were repeatedly read by SK and KN and data uploaded to the qualitative software package NVivo v11 for coding. The data uploaded consisted of all direct participant quotes from the publications reviewed.

In phase four "how are the studies related", a grid of concepts was made from the chosen studies extracts. Each study was reviewed and concepts were identified and juxtaposed to each other. This phase forms the initial assumptions.

In phase five, "translating the studies", the themes that arose across the studies are constantly compared within each other and across accounts from participants.

Phase six involved "synthesising translations", in this phase a line of argument and a new model was constructed. The line of argument reveals hidden meaning as a whole and is greater

than the sum of any one study alone. The last phase is "expressing the synthesis" and this was done by comprehensively writing up the results for dissemination.

Stages 4–6 were completed by SK and both reviewers KN and DOS were available as reviewers throughout the process. Referenced in the finding section is an example of construction of third-order constructs from first order constructs for a set theme.

## Results

### Search results

The initial search across four databases revealed 1735 articles, two additional records were sourced * (one from the reference lists of an included study and one from Google Scholar) (Fig 1). After removal of duplicates, 1404 articles remained for title and abstract screening. 1364 articles were excluded at this stage and forty articles moved forward for full-text eligibility assessment. On full-text eligibility assessment, thirty articles were excluded. Reasons for exclusion are detailed on the PRISMA (Preferred Reporting Items of Systematic reviews and Meta-Analyses) flow diagram (Fig 1). Ten studies were included for qualitative synthesis (Table 2).

### Characteristics of included articles

The majority of the articles were from North America. Six studies were conducted in the United States of America [63, 87–91], two studies were conducted in Canada [92, 93], one study was Australian [94] and one study was conducted in the UK (United Kingdom) [95]. Eight out of ten studies were undertaken in the last 5 years.

Two studies employed mixed methods in their design [87, 90]. Two studies collected data via focus groups [63, 91]. One study used participatory research and interviews to collect data [95], One study collected data in parent-child dyad interviews [94], and four studies utilized semi-structured interviews [88, 89, 92, 93]. Half of the included studies included perspectives from trans and non-binary youth and their parents/caregivers [87, 90, 93–95], while the other half focused solely on the young persons' experience.

Nine studies clearly reported their youth's sample size and these amounted to 155 young people. One study which included families, included 65 individuals, 33 of whom were young people. Therefore 188 young people and 108 parents were included in the final analysis.

### Quality appraisal

Studies were assessed for quality using CASP scoring (S1 Table). There was moderate variance across the quality scores. However, all studies were deemed to have research value and thus all studies were included in the meta-synthesis.

The researcher-participant relationship was poor across all studies with only four studies achieving a high CASP score in research reflexivity [63, 91, 92, 95].

Of note, researchers who utilized members of the trans community to aid the facilitation of interviews, workshops, and interpretation scored higher in this area. Acknowledgement of cisgender status with LGBT (Lesbian, Gay, Bisexual, Transgender) sexual orientation was noted in one paper [95]. While all studies had a clear research aim and a strong indication for qualitative research methods, some papers failed to detail the justification for their chosen research design and data collection method [87–89, 91, 94]. Similar deficits were noted on nuanced outlines of the data collection plan [87, 89, 91]. While there were methodological deficits, overall, the authors felt all papers offered significant insights into an under researched topic.

## PRISMA 2009 Flow Diagram

- **Psycinfo (1010 Records identified by search)**
- **CINAHL (201 Records identified by search)**
- **EMBASE (140 Records identified by search)**
- **MEDLINE (384 Records identified by search)**

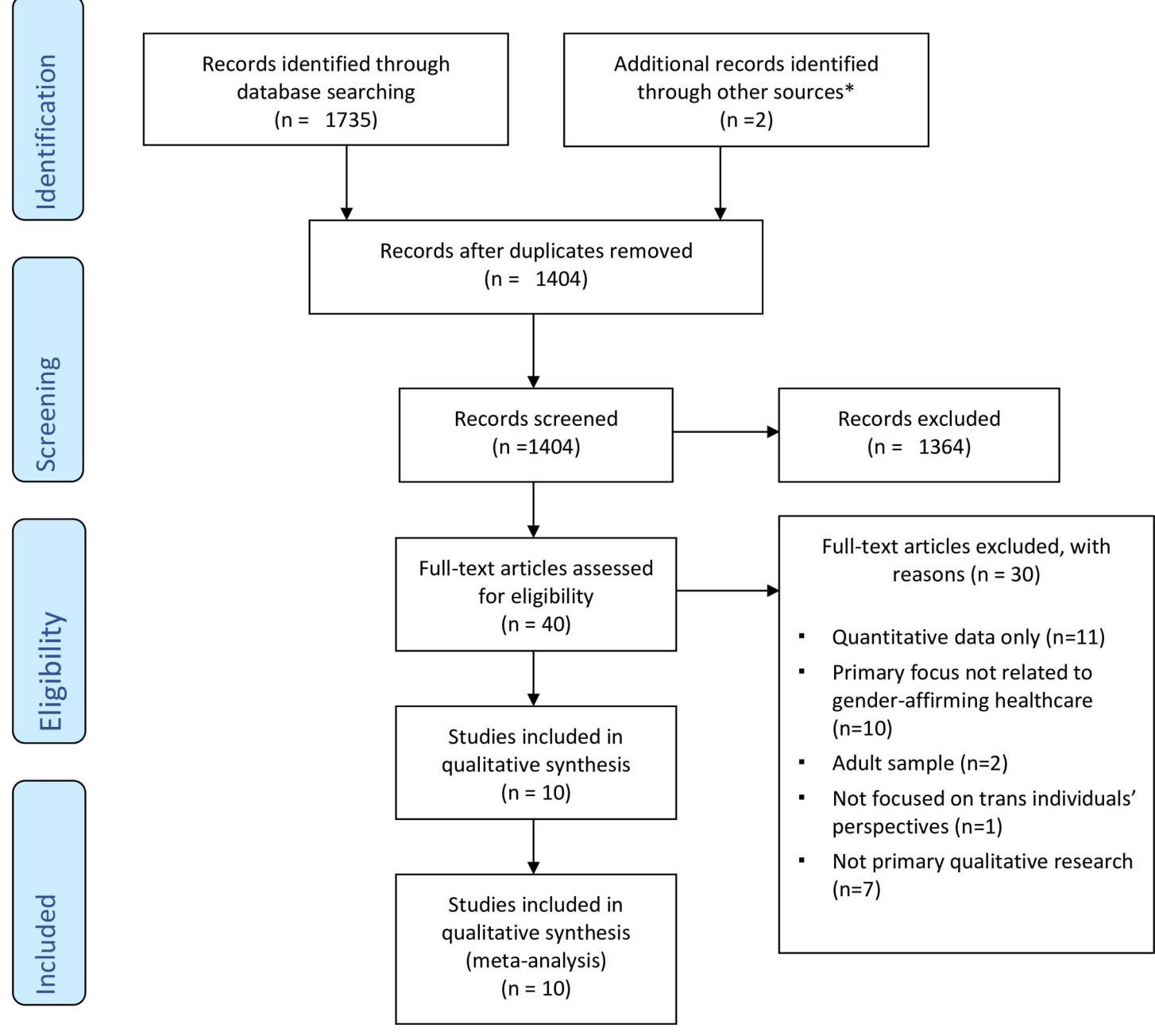

**Fig 1. PRISMA flow diagram.**

**Table 2. Characteristics of included studies.**

| Authors and year | Aim of study | Country and/ or region | Recruited from | Population/age | Sample | Age of young people | Methodology/ Study Design/ Analytic strategies | Summary of findings (themes of brief overview) |
|---|---|---|---|---|---|---|---|---|
| Breland et al. (2019) | To assess youth and parent/caregiver satisfaction with care at a paediatric multidisciplinary gender clinic. | USA; Washington | Multidisciplinary Gender Clinic | Transgender/ gender non-conforming youth and parents/ caregivers | Survey: n = 33 (TGNC youth) n = 29 (parents/ caregivers) Interviews: n = 10 (TGNC youth) n = 10 (parents/ caregivers) | 8–22 | Mixed methods: Survey and individual interviews Analysis: Thematic | Qualitatively, main themes: 1. Affirmation due to use of preferred name/ pronoun 2. Access barriers due to scheduling and readiness assessment 3. Positive interactions with a care navigator |
| Clark et al. (2020) | To explore how transgender (trans youth) and parents of trans youth made decisions around hormone therapy initiation as well as trans youths experiences of barriers to care. | Canada: British Columbia | Multiple Sources (Health Clinics/ Community Organizations/ Support Groups) | Trans youth and parents of trans youth | n = 21 (Trans youth) n = 15 (Parents of trans youth) | 14–18 | Qualitative: semi-structured interviews Analysis: Grounded Theory | Qualitatively, main themes extracted by the author: 1. Discovery, interaction and reflection practices in youth decision making 2. Discovery, interaction and reflection practices in parental decision making 3. Internalized stigma barriers 4. Systemic barriers 5. Parental support as a barrier |
| Carlile. (2019) | To elicit the experiences of transgender and non-binary children in a support group in England and investigate their experiences of healthcare provision and identify areas for improvement. | UK: London | Family Support Group | Transgender and non-binary young people and their parents | 65 people across 27 families (half adult, half youth approx.) | 12–18 | Qualitative: group workshops Analysis: Participatory Research– Illuminate model | Qualitatively, main themes: 1. Professionals' perceived lack of clinical and therapeutic knowledge 2. Mental distress caused by excessive waiting lists 3. Professionals' stereotyped gender assumptions 4. Direct discrimination within healthcare service 5. Lack of attention to parent and child voice (esp. in schools and autism realms) |
| Corliss et al. (2007) | To examine the experiences of service utilization among male to female transgender youth. | USA: Los Angeles | Trans Specific Health Clinic | MTF Transgender youth | n = 18 | 16–24 | Qualitative: semi-structured interviews Analysis: Thematic content analysis | Qualitatively, main themes extracted by author: 1. Usefulness of services (Physical, Endocrine, Mental health, Legal) 2. Barriers to care (age, financial, transport, knowledge) 3. Black market access 4. Improvement of services 5. Educating larger society |

*(Continued)*

**Table 2.** (*Continued*)

| Authors and year | Aim of study | Country and/ or region | Recruited from | Population/age | Sample | Age of young people | Methodology/ Study Design/ Analytic strategies | Summary of findings (themes of brief overview) |
|---|---|---|---|---|---|---|---|---|
| Eisenberg et al. (2019) | To describe TGD adolescents' experiences, concerns, and needs in healthcare setting. | USA: Minnesota | Multiple Sources (Clinics/ LGBTQ organizations/ Social Media/ Community events/ Referral) | Transgender identified adolescents | n = 12 | 14–17 | Qualitative: semi-structured interviews<br><br>Analysis: Inductive thematic analysis | Qualitatively, main themes: 1. Asking about gender and pronouns 2. Training for healthcare providers |
| Gridley et al. (2016) | To understand the barriers that transgender youth and their caregivers face in accessing gender-affirming health care. | USA: Washington | Multiple Sources (Children's Hospital, Trans specific clinic, Parent/support group/ Blog) | Transgender youths and their caregivers | n = 15 (youth)<br><br>n = 50 (caregivers) | 14–22 | Mixed-methods– individual interview, focus group, online survey (caregivers only)<br><br>Analysis: Thematic Analysis | Qualitatively, main themes: 1. Mandatory training on gender-affirming health care and cultural humility for providers/ staff 2. Development of protocols for the care of young transgender patients, as well as roadmaps for families 3. Asking and recording of chosen name/ pronoun 4. Increased number of multidisciplinary gender clinics 5. Providing cross-sex hormones at an age that permits peer congruent development 6. Designating a navigator for transgender patients in a clinic |
| Riggs et al. (2019) | To explore the views of Australian transgender young people and their parents with regards to medical treatment | Australia-South Australia or Victoria | Multiple Sources (Clinics/Flyers) | Transgender youth and their caregivers | n = 10 (TGNC youth) | 11–17 | Qualitative: interviews in parent-child dyads<br><br>Analysis: Thematic Analysis | Qualitatively, main themes: 1. The importance of a strong supportive parent-child relationship 2. Negotiating hormone blockers and their meanings 3. Negotiating hormones and their meanings |
| Sansfacon et al. (2019) | To develop a deeper understanding of the experiences of trans youth seeking and receiving gender care at Canadian specialty clinics. | Canada-Quebec, Ottawa, Manitoba | Trans Specific Health Clinic | Transgender youth | n = 35 | 9–17 | Qualitative: semi-structured interviews<br><br>Analysis: Grounded Theory | Qualitatively, main themes: 1. A long winding, and complicated path to access care 2. Desired medical interventions and expectations 3. Outcomes of medical interventions 4. Overall experiences with clinic care and services received |

(*Continued*)

**Table 2.** (Continued)

| Authors and year | Aim of study | Country and/ or region | Recruited from | Population/age | Sample | Age of young people | Methodology/ Study Design/ Analytic strategies | Summary of findings (themes of brief overview) |
|---|---|---|---|---|---|---|---|---|
| Sperber et al. (2005) | Identify the healthcare needs of transgender and transsexual individuals | USA- Boston | Multiple Sources (Networking within trans. Community and health care providers who have trans clients) | Transgender individuals | N = 34 | <25 | Qualitative: Focus groups | Qualitatively, main themes: 1. Disclosure of TG/TS Identity 2. Health Issues 3. Hormone Therapy 4. Mental Health Issues 5. Standard of care 6. Accessing Health Care 7. Relationships with Health Care Providers 8. Participant Recommendations |
| | | | | MTF adults | N = 9 MTF youth | | | |
| | | | | FTM adults | | | | |
| | | | | MTF youth | | | | |
| | | | | FTM youth | N = 5 FTM youth | | Analysis: not actively named | |
| Turban et al. (2017) | To identify ten recommendations that transgender and gender-nonconforming youth want their doctors to know | USA- New England | Yale Paediatric Gender Program | Transgender youth | N = 20 | 13–18 | Qualitative:Focus groups | Qualitatively, main suggestions: 1. Sexuality and gender are two different things. Totally separate 2. Talking to strangers about these things is uncomfortable 3. Nonbinary people exist 4. Names, pronouns and gender markers are important 5. Don't ask about my genitals unless medically necessary 6. Genital and breast exams are uncomfortable for most people, and they can be particularly uncomfortable for me 7. Puberty blockers and cross-sex hormones can save my life 8. Please train your staff as well. Many of us have had visits starting with the wrong tone, starting with check-in 9. If I am depressed or anxious, it is likely not because I have issues with my gender identity, but because everyone else does 10. Let me know that you are on my team |
| | | | | | | | Analysis: not actively named | |

## Gender identity interpretation

As previously mentioned, changing demographics have been noted in the population presenting to gender-affirming services globally. From the data included in this review, the author tabulated gender identity as self-reported by participants (S2 Table). Missing data is omitted with explanatory comments where needed.

Interestingly, eight papers over the last fifteen years report that transfeminine/female identities are more prevalent than transmasculine/male identities. Papers published over the last 5 years show a clear shift towards a greater prevalence of people with transmasculine/male identities. Carlile (2019) noted that participants assigned female at birth accounted for two-thirds of their young participants. While limitations exist due to missing data, small sample sizes and a North American bias, this observation is consistent with international trends [16].

## Synthesis

The research team identified 141 first-and second-order constructs across the studies, which were then interpreted into third order constructs contextualized into five dimensions:

a.  Disclosure of gender identity;

b.  The pursuit of care;

c.  The cost of care;

d.  Complex family/caregiver dynamics and

e.  Patient-Provider Relationships.

First order constructs were the direct quotes from the papers reviewed. Second order constructs were the papers original authors' views and interpretations of the data from these papers and third order constructs were the views and interpretations of our synthesis team expressed as key concepts.

An overview of each study's contribution to each dimension is outlined on a table (Fig 2).

A comprehensive tracked example of first to third order constructs can be found in S3 for patient-provider relationships (S3 Table).

## Findings

Each of these dimensions represent potential barriers and experiences that transgender and non-binary youth may face during healthcare navigation. These dimensions span five non-linear pillars of personal, biomedical, psychosocial, economic and environmental conditions. Each theme is discussed and direct quotes included to illustrate the findings.

**1.1. Disclosure of gender identity.** Three studies flagged reluctance to disclose gender identity as a major systematic barrier to accessing gender care [63, 92, 93]. For most young people, parental/guardian consent is a prerequisite to accessing care, and thus "coming out" is an imperative step.

*1.1.1. Postponing disclosure of gender identity*. This is evidenced by studies which detailed that young people postponed this step due to fear of not being accepted [92], fear of being bothersome [92] or a lack of comfort talking about gender questioning [93]. Even at the point where participants were receiving care, (after coming out) they worried that health care providers would document their gender identity status and this may negatively affect their insurance coverage [63].

> *"Well at first, I like, you know, I wanted blockers and I wanted hormones, and I like never talked about it. . .. So, you know, I think I probably, you know, I could've have gotten them like sooner. But at that, I was so shy, and I was like, No, I don't want to like bother them or anything" (Yannick, TM, 16 yo) (Sanfacon, 2019) [92].*

*1.1.2. Factors encouraging disclosure.* Prior to disclosing their gender identity, gender non-conforming and gender questioning youth often found solace and guidance from other trans youth. This was noted as peer-peer support and online via sites such as YouTube. *"One thing that I really did find helpful was on YouTube they actually have trans people who will go on and talk about their own experiences"(Clark, 2020)* [93]. This also helped to set expectations for what barriers may exist going forward seeking gender-affirming care.

**1.2. The pursuit of care.**   After coming out and progressing to the steps of finding a healthcare provider, youth face major structural barriers. This is evidenced in the literature as finding a suitable health care provider [63, 90, 93], geographical burdens [63, 93], and onerous waiting lists [87, 90, 92, 94, 95].

*1.2.1. Finding a provider.* Finding a competent health care provider that could advise young people and their families was identified as a major systematic barrier. This is evidenced by a participant in Gridley and colleague' study detailing how it was initially challenging to find a provider who was accepting new patients, who worked with adolescents, who accepted the person's insurance and who were trans-friendly [90]. Equally, mental health support that was trans specific was difficult to source for many participants.

*1.2.2. Geographical threats.* Geographical locations were identified as a major systematic barrier to accessing gender-affirming care [63, 93]. Many participants were forced to elect for providers that were based far from their homes and described this as *"really inconvenient"(Sperber, 2005)* [63]. The urban-rural divide was particularly evident, with one young person travelling from a rural area to Boston to receive care [63]. Participants knew *"people that live in [town] that have to travel 8 h to see the people that they need to, which is incredibly expensive and not accessible to all"(Clark, 2020)* [93].

*1.2.3. Onerous waiting times.* Waiting times surface as the most prominent systematic barrier in accessing gender-affirming care [87, 90, 92, 94, 95]. A parent noted that the waiting times are unfortunate and contributes towards the dysphoria and anxiety that impacts their child and family unit negatively [93]. Carlile details challenges faced in the UK, having to wait for the Child and Adolescent Mental Health Service review prior to being referred for the Gender Identity Development Service. Both of which have independent waiting lists.

A youth in Inward-Breland's study highlighted that once in the service, the wait does not always end, and multiple appointments might remain before starting hormones. The lack of communication around this is reported as frustrating. For many young participants, it is not only the arranged and printed appointment time that appear on an outpatient letter but frequently the extensive waiting period over years and the multiple barriers they had to overcome to get to this one. *[The process was] way too long. "Cause I feel like I've been waiting for this transition since I was 6 years old" (Steve, TM, 17 yo) (Sanfacon, 2019)* [92]. Throughout the wait, parents are often waiting for a cancellation or an opportunity to reduce the waiting time.

**1.3. The cost of care.**   Financial barriers may present challenges for trans youth and their families.

*1.3.1. Guilt and worry.* Cost and insurance coverage are repeatedly identified as major systematic barriers to care [63, 90, 92, 93]. This is evidenced in the literature by young people who worry about how much it will cost and if their insurance will cover the cost. Some young people felt guilt and worry that their parents were burdened *"And hopefully, you know, when I get a job, hopefully, I can take that off their backs, and I can do that for myself" (Jake, TM, 16 yo) (Sanfacon, 2019)* [92].

*1.3.2 Disparity in insurance plans.* For those, not in school or work, insurance can be difficult to obtain [63]. Frequently, individuals will choose a specific specialist first and due to their expertise or trans-friendly reputation, but then they find out that they may not be covered by their insurance. Participants noted that disparity exists in which insurance plans cover what.

| Thematic Synthesis | | | | | |
|---|---|---|---|---|---|
| Authors | Theme 1: Disclosure of Identity | Theme 2: Pursuit of Care | Theme 3: Cost of Care | Theme 4: Complex Family/Caregiver Dynamics | Theme 5: Patient Provider Relationships |
| Breland et al. (2019) | | X | | X | X |
| Clark et al. (2020) | X | X | X | X | X |
| Carlile (2019) | | X | | X | X |
| Corliss et al. (2007) | | | | | X |
| Eisenberg et al. (2019) | | | | | X |
| Gridley et al. (2016) | | X | X | X | X |
| Riggs et al. (2019) | | X | | X | |
| Sansfacon et al. (2019) | X | X | X | X | |
| Sperber et al. (2005) | X | X | X | | X |
| Turban et al. (2017) | | | | | X |

**Fig 2. Contribution of included studies towards themes.**

In some instance's parents/caregivers must battle just to have hormones or surgeries covered [90].

This is a barrier that can be addressed by flexible insurance plans. Flexible insurance plans and health care providers who were invested in gender care, were identified as enablers that enhanced access to gender affirming interventions and helped to overcome some of the structural and financial barriers.

**1.4. Complex family/caregiver dynamics.** Across six studies, families and caregivers emerged as key stakeholders impacting access to gender care [87, 90, 92–95]. Only one of these six studies did not include parents/caregivers in their sample [92].

*1.4.1. United front.* Families/Caregivers as enablers to accessing healthcare was identified as a major theme [87, 90, 92–95]. Families'/Caregivers' willingness to listen and learn facilitated the process of accessing gender-affirming care. This was evidenced by several studies where parent and child were detailed as navigating the gender journey together as a united front.

*"I remember my mom and I talking about what, next steps. At this point, I was still very unsure, and so my mom had the idea that I could talk to someone. [. . .] Maybe so they could help me, guide me in the track that I'm trying to get. And so she found the [specialty clinic] here, and so I came here and I met with [intake nurse]. That was awesome" (Adrian, TM, 14 yo) (Sanfacon, 2019)* [92].

*family support is of paramount importance to the young trans person for acceptance, help, support, and friendship. All need to be united and well-informed" (Carlile, 2019)* [95].

*1.4.2. Insurmountable barrier.* Conversely, Families/Caregivers can be major barriers to accessing healthcare. This is evidenced by half of the referenced studies, in which parents/caregivers are portrayed as inhibitors to gender care. Two studies noted the protection of a child's fertility and a wish to have biological grandchildren as a factor taken into consideration [90, 94]. Studies also reported incidences where families were reluctant to support or had reservations about a young person's gender incongruence. This is evidenced by the experience of one parent who was supportive of their child's gender identity but was averse to any medical intervention [93].

Another parent interviewed by Gridley et al was not entirely convinced of their child's gender identity and were using "insurance exclusions" as a stalling tactic [90] as they felt time may change their child's desire to access gender affirming care. To access gender-affirming care in the form of GnRH blockers, hormones, surgery, or even psychosocial support, parental support is a prerequisite, and a sizable barrier if missing.

*1.4.3. Advocates.* Upon acceptance of gender identity, half the studies report that parents/caregivers become fierce advocates for their children [90, 92, 94]. They take on administrative challenges, schools, health systems, and insurance firms. Young people acknowledge the positive role that their parents play in battling health systems that perpetuate barriers. Parents/caregivers often take on the advocate role and serve as a catalyst for accessing gender-affirming care. Of note, little exploration of the motivating factors for change in acceptance or youth perceptions of this is documented in the literature.

*1.4.4 Family as "patients too".* Other experience noted were from several studies which indicated that families/caregivers have unique personal needs throughout the process [93–95]. One father described his family as "patients too" and elucidated the mental health and family pressures that exist through their child's gender journey [95]. Families noted that healthcare professionals can provide invaluable reassurance and parent-parent support can be comforting [93]. Overall, parents go through a complicated educational process learning when their child comes out, and they often have to learn in a pressurized environment [94]. Literature scarcely mentions the impact on siblings and the supports they may need in a family unit.

*"One of the workshop parent participants recounted a difficult family weekend away where her trans child was "trying to stab himself with a knife" and his cisgender sibling also cut her arms with a piece of plastic. She said that when they returned home, she just felt flattened."* *(Sanfacon, 2019)* [95].

**1.5. Patient-provider relationships.**   Throughout the studies included in this review, variance exists on how youth access gender-affirming care. In some instances, young people present to a specialised gender clinic and all gender-affirming assessments are carried out in this service by a specialised multidisciplinary team (MDT) [87, 88, 91, 92]. Others find a general physician who will provide gender-affirming care without MDT support. Five of the studies included in our synthesis enrolled patients who all accessed their gender affirming healthcare from different sources [63, 89, 90, 93–95].

*1.5.1. Ill-equipped.* Reports of healthcare providers (HCPs) being ill-equipped on gender-related care were common throughout the studies [63, 89–91, 95]. For Eisenberg's client, participants expressed a wish that HCPs would educate themselves on the basic forms of medical transition and knowing what procedures exist and how to access them [89]. In Carliles' study a parent described how the Child and Adolescent Mental Health Service had little trans knowledge and their child's anxiety remained untreated [95].

In many cases, youth and parents became aware that the HCPs were ill-equipped to provide them with adequate care because the HCPs themselves stated that they were not proficient or comfortable in providing gender care [90]. In other cases, people found themselves being referred between therapists as they felt it wasn't their area of expertise [63], or being told their prescriptions for gender affirming hormone therapy are on hold until therapeutic guidelines are found (presumably due to a lack of confidence around continuing the hormone therapy) [63]. Often information was shared by the HCP that was deemed dense, complicated and hard to understand from the young person and/or parent's perspective [90].

*1.5.2 Dread, fear and avoidance.* Due to HCPs lack of confidence in dealing with trans patients, many young people avoided HCPs for general health concerns. Participant's detail the feelings of dread and fear of navigating these spaces. *"I'm deathly sick, but I don't want to have to go try to deal with all the hate at the doctor's office" (Eisenberg, 2019)* [89]..…*"I have to be deathly sick before I go to a doctor" (Sperber, 2005)* [63]. For those who make it to the doctors premises, it has been noted that all staff require sufficient training on gender, as it is not just the interaction with the HCP but with the receptionist and all other support staff. *"I can't even make it through the front door without staff staring at me, laughing at me or whispering about my gender presentation" (Sperber, 2005)* [63].

*1.5.3. Prove gender identity.* Distressingly, many young people experienced instances where they felt obligated to prove their gender identity [63, 88, 90, 93, 95]. Proving gender incongruence also demanded a trans binary narrative with very firm notions on the expectations of masculinity and femininity through gender expression. One young person came to an appointment wearing jeans and trainers and was told she wasn't taking it serious enough [95], another still liked to paint his nails and he was told he wasn't masculine enough [90] and many participants felt they had to prove that they were "trans enough" to be cleared for gender-affirming treatments [90].

One young person described it as follows: *"It's like being stuck in a wet, cold, sandy, uncomfortable swimsuit that is too tight, and everybody else is wearing warm, dry clothes . . . and somebody not letting you have [warm dry clothes] until you prove [your gender]. It's rather irritating and uncomfortable and angry-making and depressing and all that other evil, nasty stuff" (Clark, 2020)* [93].

*1.5.4. Pronoun/name etiquette.* Using the wrong pronouns and using a name that is not preferred was common across the studies [90, 91, 95]. This left young people feeling invalidated and judged. One caregiver on describing the loss of a [male] child and embracing a new [female] one, equated a misgendering appointment reminder to "a dagger in the heart" *(Gridley, 2016)* [90]. Responses to mis-pronouning varied from young people feeling discomfort and that all eyes were looking at them to a shouting match between provider and patient. *"the doctor said, 'her, her, her' and [my son], who's 10, said, 'him, him, him!' and the doctor got mad and started sort of being dismissive and irritated, and kept saying 'her!' this weird sort of oppositional like 'I'm not calling you that'" (Gridley, 2016)* [90].

Good etiquette on asking someone's pronoun was explored in multiple studies. In the study by Eisenberg et al a participant shared that *"even if they say something that isn't right or just kind of makes me feel uncomfortable, it's still good for them to ask, because I can always just correct them, and, if they really care, then they'll listen to me (Participant 2)" (Eisenberg, 2019)* [89].

Young adults from Yale Paediatric Gender Program had similar advice for HCPs or example: *"Hello, my name is Dr. Jones, and my preferred pronoun is she/her. What is your preferred name and pronoun?" I may in turn suggest that my preferred pronoun is "he,", "she," or "they," among other possibilities. Remember that it is always better to ask than to assume" (Turban, 2017)* [91].

*1.5.5. Refusal of care.* Some individuals have been refused healthcare due to their gender identity. *"I was actually turned away [from an emergency room] because the doctor said he did not treat people like me," said one FTM youth (Sperber, 2005)* [63]. Reports of discrimination of this nature is not uncommon across the literature. Commonly, youth report that they were refused access to gender-affirming services [88, 90, 95]. Youth were turned down due to their age [88], due to their gender expression [90], and were blocked through legal restrictions [88].

This is understandably experienced as gatekeeping of care. HCPs that act as gatekeepers are usually primary physicians for whom gender care is not their areas of expertise. Refusal of referral has led to circumstances where young people seek and obtain hormones without prescription [63, 88, 90]. They described feeling desperate and were aware that *"you could get seriously hurt by doing it" (Gridley, 2016)* [90] but they didn't want to *"wait around six months so someone could tell me, 'you are what you say you are.'"(Sperber, 2005)* [63]. One young person outlined their accidental overdosing and how this could have been avoided [88].

*1.5.6. Positive experiences.* Positive experiences with HCPs and health services were also noted [87, 88, 90, 92]. One participant in Corliss et al's study describes how engagement with healthcare services made her feel more like a woman and helped her to grow and find herself. Inward-Breland speak positively about the role of a care navigator in the gender service who assisted with paperwork, legal changes and all official documents needed. Specialist clinics received more positive praise than primary care services. Two studies commented on the positive impact that a nurse has on the service [63, 92].

## Line of argument synthesis: The rainbow brick road

Noblit & Hare suggested that a "line of argument" synthesis entails the construction of an interpretation from multiple bodies of work. It involves revealing what is hidden in the studies and bringing it together as a cohesive summation.

Applying this principle to the current meta-ethnography the authors devised a model coined "The Rainbow Brick Road" (Fig 3). This non-linear road represents reciprocally translated dimensional obstacles that transgender and non-binary youth may experience from their initial gender questioning through their healthcare navigation. Applying the story of the Wizard of Oz as a metaphor to the journey of obtaining gender-affirming care as a young person. The Lion who desires courage represents the courage needed to disclose a transgender/non-binary gender identity to oneself and others. It also represents the tenacity and ferocity needed to overcome structural, environmental and economic barriers that arise The Tinman who desires a heart represents the acceptance of family/caregivers who may then act as fierce advocates or insurmountable barriers. The Scarecrow who desires a brain represents healthcare providers who are ill-informed and received little training on gender-related care.

Each road is unique for the young person on the journey and this journey is a spherical one, not linear. Each dimension identified represents personal, biomedical, psychosocial, economic, and environmental conditions for youth and acts as a map to navigate the journey and plan for the road ahead. While the "Rainbow Brick Road" is primarily from the perspective of the young person, each of the dimensions/pillars identified prompts greater action needed from community and from health systems. From a personal perspective, trans youth need local LGBTQ support groups and trans inclusive school policies to help support the process of "coming out" and exploring ones' identity. Environmental factors can be remediated by having the availability of

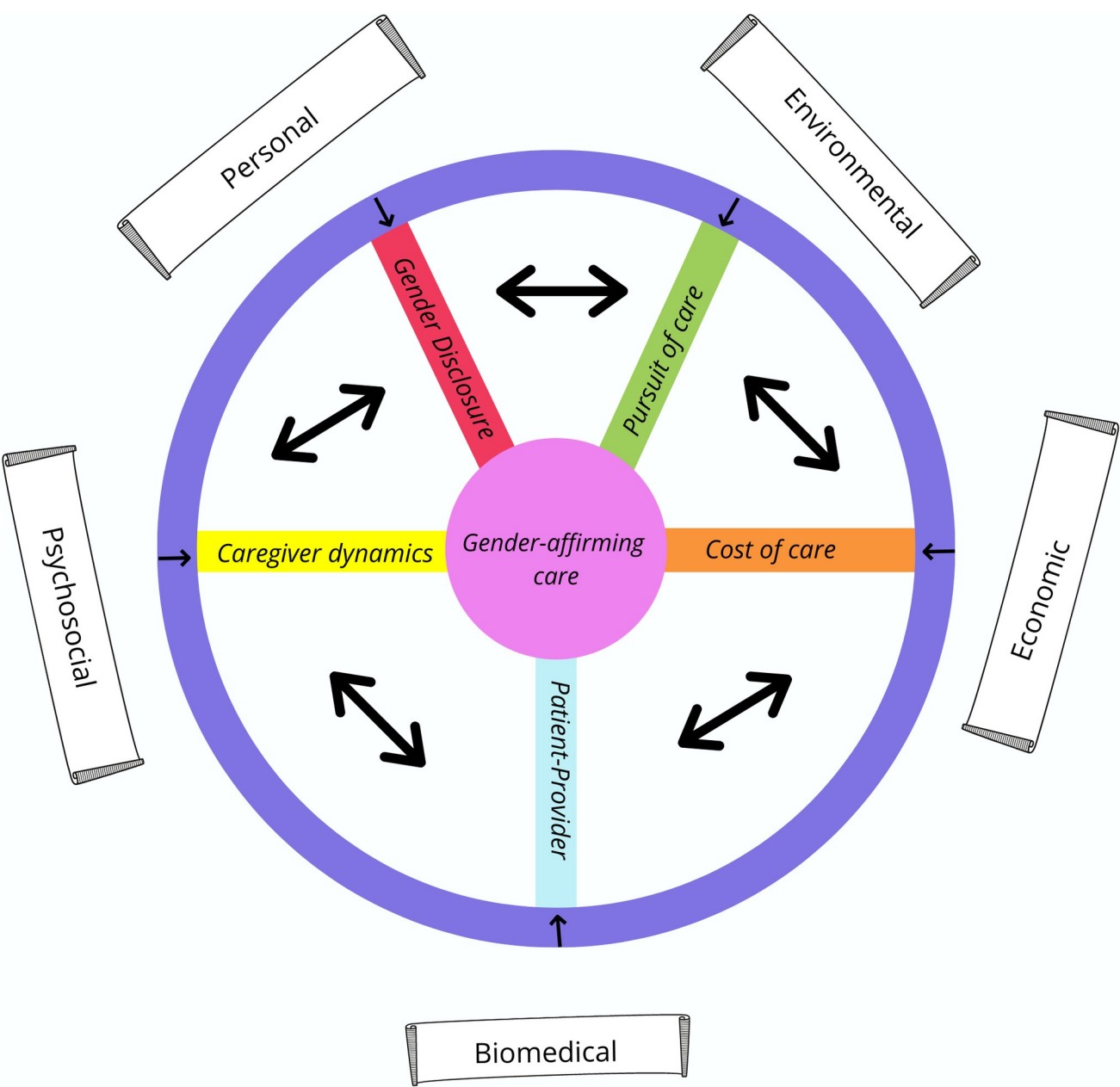

**Fig 3. The rainbow brick road model.**

trans specialist services locally and healthcare navigators to assist along the way. Economic factors call for the public provision of gender-affirming care and trans inclusive insurance policies.

From a psychosocial end, the complex family factors demand more family support/therapy at a community level and biomedically, trans youth require competent healthcare professionals who have specialist training on gender related care and healthcare navigation.

The "Rainbow Brick Road" serves to identify some of the non-linear paths that youth may traverse, describe personal attributes needed by multiple stakeholders through characterisation and suggest additional bricks needed to support the larger dimensional pillars identified.

## Discussion

This systematic review and meta-ethnographic study synthesized all the available qualitative studies around the experiences of transgender and non-binary youth accessing gender-affirming care. Ten studies were included, all published since 2005, capturing 188 young people and 108 parents' voices.

To the authors' knowledge, this is the first qualitative systematic review on this topic and serves to highlight the dearth of qualitative evidence detailing the experiences of transgender and non-binary youth accessing gender-affirming care. As per interpretation from the research team, five dimensions describe factors that influence healthcare accessibility and satisfaction. A young person's experience is shaped by a.) disclosure of gender identity; b.) the pursuit of care; c.) the cost of care; d.) complex family/caregivers dynamics; e.) patient-provider relationships.

Our primary findings relate to a dynamic set of factors that describe the experience of transgender and non-binary youth in obtaining access to gender-affirming care. One of the insights arising from the included studies was how different factors called for further needs from health systems and the community on a whole. Generally, it is well established that trans and non-binary youth need a safe space to explore their gender identity and have access to adult role models and healthcare professionals that affirm the exploration without influencing the final outcome. Secondary to allowing space to explore gender identity, youth need access to general healthcare that is respectful of all gender identities, access to expert gender-affirming care if that route is recommended, and continual support and access to mental health and psychosocial supports.

As per the included studies, to enable positive outcomes, youth require access to unbiased and accurate medical information, sufficient insurance and income, support from family and caregivers, and support from healthcare professionals well-trained on gender-related issues.

Our model identified certain health beliefs that are evident in the trans community. Expectations of stigma and discrimination are recurrent themes across qualitative and quantitative research [96]. A hegemonic transnormative, trans-binary presentation is perceived to be the expectation that one must meet on their gender journey. This is evidenced in the findings by the sub theme of "needing to prove your gender".

Among some members of the transgender community, there is a perceived health belief that medical or surgical intervention is a requirement to legitimise the trans experience. This has the potential to minimise the role of healthcare professionals in providing care for trans individuals who do not seek medical or surgical intervention but may seek social transition support or psychosocial supports. This focus on medical aspects of transition can also inadvertently devalue the affirming power of personal and social transition. It is striking, but unsurprising, that the focus for young people is hormones, and that barriers are perceived as barriers to hormones rather than barriers to care. Specialist gender services may provide supports such as speech and language therapy, assistance with legal document change and paperwork, peer-support groups, family counselling or support, family planning services and any of these services have the potential to be gender-affirming.

Our line of argument theory, the Rainbow Brick Road, outlines the multiple factors identified as important in the experience of young people accessing gender affirming healthcare in the current evidence base. However, it is important to acknowledge the paucity of data and that there may be additional important factors that are as of yet unidentified. In addition, "the Rainbow Brick Road" is unique for every young person.

Miriam Ryvickers' behavioural ecological model compliments our line of argument theory and enables the categorization of the potential trans-specific healthcare navigation factors that

affect the experience of young people accessing gender affirming healthcare. The authors' findings, as represented by the Rainbow Brick Road model, can be translated to Ryvickers' model and elucidates the unique factors specific to transgender and non-binary youth (Fig 4).

Ryvicker derived a behavioural-ecological framework in order to help to describe healthcare navigation as access to healthcare becomes more convoluted and individuals require more advanced skillsets to achieve care [97]. This framework merges and builds upon two existing models, Ronald Andersens' healthcare model [98, 99]and Satariono's epidemiology of aging [100]. Healthcare navigation is understood as a process (or processes) that individuals and/or their caregivers negotiate with regards to healthcare needs, constraints and outcomes [97, 101].

Ryvicker's model posits that healthcare navigation is an "ecologically informed process" not just based solely on the spatial distribution of services but also due to individual and environmental factors that influence decision-making.

Individual factors such as age, race, education, religion, income, political stance, and geography are important factors in healthcare navigation. Specific to the trans community, as outlined in our findings, the emergence of transgender identity and the disclosure of transgender identity to others greatly influences healthcare utilisation. Lev's theory of transgender emergence relates to the findings of our theme "disclosure of gender identity" [102]. Lev's model documents various personal stages a trans person may navigate to achieve integration (self-acceptance). Awareness, seeking information, disclosure to significant others, exploration (of transition related needs and personal labels), and lastly integration. In our findings, awareness, seeking information, disclosure to significant others, and exploration all arose as key themes.

In addition, health literacy has been identified as a predisposing factor in our analysis. Many of the studies included in our analysis alluded to challenges with comprehension of medical information. Healthcare literacy is an example of an enabling factor: a resource that enables and facilitates service use or healthcare access. As per our analysis, access to accurate

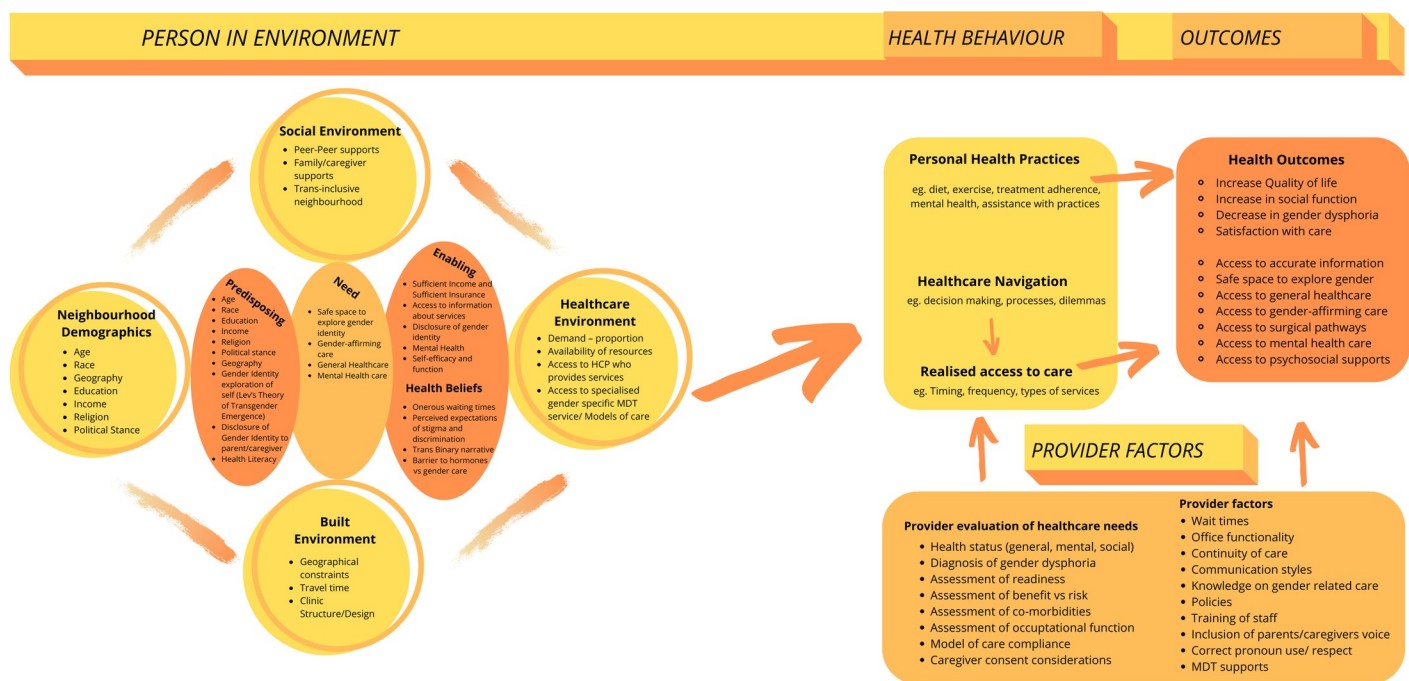

**Fig 4. Adapted Ryvickers' behavioural ecological model for healthcare navigation of transgender and non-binary youth.**

information that is easily understood is an enabler to healthcare. In number of studies included in our analysis, young people and their parents found healthcare information provided by healthcare professionals to be difficult to understand.

Other enabling factors include sufficient income and adequate insurance, as detailed in our theme "the cost of care". Specific to the transgender community, and evidenced by our model, the privilege of being able to disclose one's gender identity and be supported is a key enabling factor. Without this, access to care as a young person may be impossible.

A factor not clearly identified in the studies included in our analysis, but a key enabling factor in clinical practice, is an individual's self-efficacy skills. Impaired self-efficacy can be compounded by poor mental health, poor social health, and neurodiversity. Self-efficacy with regards to healthcare navigation and personal health practices is an area that warrants further investigation and research.

While recognising that individual factors influence access to care [98, 99], Ryvicker's model also proposes that individual factors are only one component to understanding enabling and constraining factors involved in healthcare utilisation. Neighbourhood demographics, healthcare environments, social environment, and provider factors all play an instrumental role in healthcare navigation, realized access, and therefore healthcare outcomes.

Ryvicker's model notes that neighbourhood demographics influence health behaviour and outcomes. The age, race, education, income, religion, and political stance of the people in your surrounding area can influence your own inherent views, and influence the confidence one might have in receiving support if deviating from the local societal norm.

For youth, these neighbourhood demographics may influence the parent/caregiver views whose support is required to navigate care. The geographical area, that encompasses an individuals' home contains unique environmental characteristics which may act as variables in healthcare navigation, be they physical, social or economic. In our analysis, physical distance from healthcare providers was a dominant theme. Given that the current prevalence of transgender individuals within a population is relatively low, the likelihood of having access to specialised services is dependent on where you are living. As outlined in our findings, individuals often have to travel long distances to access care, more often located in an urban area that has more socio-economic resources. This links strongly with our third order construct the "pursuit of care", which highlights the difficulty in finding a provider and the transport and geographical challenges. Ideally, healthcare should be accessible from a reasonable distance, and failing this, the use of technology or medical partnerships and shared care agreements should be considered to overcome this critical barrier to care.

As well as physical factors, our analysis noted several inferences to effects related to economic factors (related to insurance coverage) and social factors. With regards to the healthcare environment there are several economic factors that influence healthcare access.

Challenges posed by insurance coverage, and cost of care, were was key themes identified through our synthesis. Of note, most of the studies included in our analysis were completed in countries where healthcare provision is primarily in the private sector and dependent on insurance coverage. Disparities of coverage were detailed, with some plans being trans-exclusionary. Youth often felt burdened by the financial risk and the insurance discrimination that they may face; as outlined in our findings under "the cost of care". The socioeconomic resources of individuals within a geographical area and supply of specialised providers also constitute access challenges. Interestingly, we did not find any research that explored accessibility in systems with two-tiered healthcare (private and public options) or additional financial factors and barriers perceived by individuals in healthcare systems where provision is primary via the public sector.

Social environment has a clear influence on healthcare outcomes. Communities with high perceived social capital and a functional interpersonal space with shared norms, values, and understandings typically have more positive self-perceived health and positive health-behaviours. This is linked to neighbourhood demographics and economic status.

For the trans community social capital could be an enabling or constraining factor depending on level of education and allyship to transgender communities. For trans youth, as identified in our findings, family and caregivers can become fierce advocates or insurmountable barriers. The influences of peer support, trans-supportive groups, and school systems are under-explored social environment factors and have the potential to help people with their personal and social transition. As waiting times increase, there is a distinct opportunity to explore what sort of services can be offered and availed off to potentially alleviate gender dysphoria while awaiting review in a healthcare setting. Socially transitioning prior to engaging with healthcare services could prove to be gender-affirming and if a young person has socially transitioned, and is accepted in their community, this could have a significant positive influence on their health.

Many provider factors influence access to healthcare. These factors include wait time, office functionality, continuity of care, communication styles, and respect. For the trans youth community, the provider-patient relationship had the potential to be an enabling force on the gender journey or a constraining factor that is experienced as gatekeeping. The provider-patient relationship is an important recurring theme in our analysis. Generally, across the identified studies it was found that healthcare professionals need to enhance their knowledge of gender-related issues, implement trans-inclusive policies, improve staff training, include patent/caregiver perspectives, use correct pronouns, and involve relevant MDT supports. This has the potential to positively influence patient and family satisfaction with care, ultimately improving individual outcomes. It is noted that ethical dilemmas can exist, where complex family structures are apparent and where youth are under the age of consent. More work is needed to understand and address these ethical dilemmas and to offer clearer guidelines to support the provider-patient relationship in these circumstances [103].

While the patient perception of need is important, so too is the provider evaluation of healthcare need. This can have a profound impact on healthcare outcomes as providers have a key decision-making role in healthcare provision. Healthcare providers have a duty of care and ethical responsibility to assess health status (including physical, mental, and social health) and to use this assessment to determine the risk benefit balance before recommending any clinical intervention. Co-morbidities and/or potential complicating factors need to be assessed. Any identified risks need to be addressed, and remediated as much as possible, so that the potential therapeutic benefit of a clinical intervention outweighs the identified risks.

Providers often must work within the confines of clinical guidelines, models of care, and local medicolegal guidance, which all have their own requirements and pre-requisites. Commonly, this will include assessing for or obtaining proof of a diagnosis of gender dysphoria. With regards to youth under the age of consent, caregiver consent will likely be a legal requirement. As outlined in our theme of patient-provider relationships, these evaluations of need can be complex and require a sophisticated skillset. Often healthcare providers feel ill-equipped to provide safe and effective care. Naturally, any assessment of need by a provider can feel uncomfortable and can be experienced by a young person as a threat to their gender identity, further evidencing the need for training and specialism to facilitate this in an affirming and positive way.

Individual and environmental factors greatly influence health behaviour (Fig 4). Health behaviour under the model includes personal healthcare practices, healthcare navigation, and ultimately realized access to care. Healthcare practice represents personal compliance with

things such as diet, exercise, treatment adherence, and mental health management. For youth, assistance with these personal practices is another factor for consideration. Healthcare navigation pertains to the processes, decisions, and conflicts that influence access to care, which can intersect with provider factors to help or hinder health outcomes.

Health outcomes in this population can be individualised depending on the perceived needs of a person. Generally, outcomes would include an increase in quality of life, decrease in gender dysphoria, increase in social function and overall satisfaction with the level of care received. In all, it should mean that individual needs are met and evaluated positively by providers resulting in access to accurate information, a safe place to explore gender, and positive healthcare outcomes.

Findings from our meta-ethnography and our model clearly shows the complex factors that influence healthcare utilisation for transgender and non-binary young people. We have presented a novel line of argument that integrates a behavioural-ecological models of healthcare navigation and revealed some of the potential experiences and barriers one might face as a young person seeking to access gender affirming medical interventions. Further investigation is recommended to explore in more depth the spatial and ecological factors effecting youth and their caregivers as they navigate gender affirming healthcare.

## Limitations

Firstly, qualitative evidence synthesis is interpretive and therefore different researchers may derive different findings. Therefore, the rationale and tracking of first-order and second-order constructs is clearly detailed in the results section and outlines how findings were formed.

The number of studies included in this review was relatively small, which is due to the dearth in research responding to the experiences and needs of this population. From the studies included, quality varied, but overall all studies were deemed of sufficient quality to be included. The risk of selection bias and reporting bias is noted and the CASP scoring system helps to counteract this. The sample size was relatively low in all studies and the sampling techniques employed had significant potential for bias. A reliance on snowball, purposive, and convenience sampling was evident. This can result in over-representation of parents who are supportive of their child's gender journey.

Geographically, 80% of the studies were North American, one Australian, and one from the UK. As countries societal acceptance and healthcare systems have a significant impact on the experiences of accessing care, there is a distinct lack of research from outside of North America. Ultimately, while some barriers are ubiquitous, others can be country or culture specific. Similarly, healthcare provision varies significantly between countries, with resultant disparity in the experience of institutional, organizational and financial barriers.

While most studies included non-binary and gender non-conforming individuals in their sample sizes, there was a distinct lack of exploration surrounding barriers that are unique for the non-binary population. The trans binary narrative resonated across most studies and parents' understanding, experiences, or views on non-binary or gender non-conforming youth was minimal at best.

Finally, it is noted that the voice of the health care provider was not included in any studies. This deprives us of information on the healthcare providers perspective, which would be informative in defining barriers to care from an institutional and organizational level.

## Implications for practice

The third-order constructs and line of argument that were forged for this paper, expand the understanding of trans and non-binary young peoples' experiences in accessing gender-affirming care. Healthcare providers should be knowledgeable of potential challenges youth and their families may have experienced in the lead up to obtaining their care. This paper serves to map some of these experiences via the "Rainbow Brick road". Furthermore, this paper highlights that greater action is needed from both health systems society in genneral.

Often youth felt the need to present in a transnormative trans-binary manner and healthcare providers should create a safe space to discuss openly their gender identity, whether binary or otherwise. Training all staff on pronoun and name etiquette is a simple measure that can have positive impact for service users. As service needs increase, healthcare providers should seek to offer and partner with specialised services that can provide affirming support to youth such as speech and language therapy, peer support groups, family planning, counselling and any other needs a young person or their family will have.

This research may have positive implications for providing context for all stakeholders to potential experiences accessing care. The literature echoes that many individuals felt the road was unclear and long. While this research does not shorten the road, it may act as a map and help form expectations of challenges ahead and help to plan how to combat them.

## Implications for future research and policy

To the author's knowledge, A behavioural-ecological model has not been applied to young people who are transgender or non-binary in the context of accessing gender-affirming care. This lens creates a foundation for future research and exploration of factors leading to successful healthcare navigation and utilisisation. Notably exploration of neighbourhood, social and healthcare environments from a qualitative focus would be beneficial to understanding the nuances of healthcare navigation for this community

This meta-ethnography demonstrates the lack of comprehensive clarification of the unique barriers and experiences faced by non-binary and gender non-normative individuals. Further investigation to identify factors specific to non-binary youth would be beneficial and formation of best practice guidelines and policies would be useful for practitioners. WPATH and other guidelines have a dearth of information on medical or surgical interventions for non-binary individuals.

Health literacy and the search for medical information for youth and families was a recurring theme and should be explored further. Family planning in this population received very little attention qualitatively and is a target for further research.

Overall, researcher reflexivity was identified as weak but studies that employed participatory action in their methodologies were much stronger in this respect. The inclusion of trans individuals as stakeholders in research and policy formation is recommended for future research to improve reflexivity.

Lastly, the preferred model of gender care is an interesting topic of further research. Some individuals prefer access at a primary care facility whereas others may prefer specialised gender services. An investigation into models of care and assessments would be beneficial to ascertain youths preferred healthcare environments and to shape policy formation.

## Conclusion

In conclusion, this paper enhances the understanding of the experiences of transgender and non-binary youth accessing gender-affirming medical care. It is the first paper to do this through a behavioural-ecological perspective and map this to a novel line of argument

synthesized from the included papers. Personal, environmental, biomedical, economic and psychosocial barriers are named as major factors in determining if a person will be able to access gender affirming care at a young age. We further identified healthcare navigation as influenced by individual, environmental, and provider characteristics with respect to realized healthcare access and healthcare outcomes. Deeper understanding of the factors and barriers involved in healthcare navigation can help to mitigate against these and the authors have thoroughly synthesized this research with potentially positive implications for practice and further research.

## Supporting information

**S1 File. ENTREQ checklist.**
(DOCX)

**S2 File. Search strategy.**
(ODT)

**S3 File. PRISMA checklist.**
(DOCX)

**S1 Table. CASP scoring for included studies.**
(ODT)

**S2 Table. Self-reported gender identities.**
(DOCX)

**S3 Table. First-third order constructs tracked (Theme 5).** Patient-provider relationships.
(XLSX)

## Author Contributions

**Conceptualization:** Seán Kearns, Karl Neff.

**Data curation:** Seán Kearns.

**Formal analysis:** Seán Kearns, Karl Neff.

**Methodology:** Thilo Kroll, Karl Neff.

**Supervision:** Thilo Kroll, Donal O'Shea, Karl Neff.

**Writing – original draft:** Seán Kearns.

**Writing – review & editing:** Thilo Kroll, Donal O'Shea, Karl Neff.

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
