## [Decision Letter · Decision Letter 0]

14 Jul 2021

PONE-D-20-38613

Experiences of transgender and non-binary youth accessing gender-affirming care: a systematic review and meta-ethnography.

PLOS ONE

Dear Dr. Kearns,

Thank you for submitting your manuscript to PLOS ONE. After careful consideration, we feel that it has merit but does not fully meet PLOS ONE’s publication criteria as it currently stands. Therefore, we invite you to submit a revised version of the manuscript that addresses the points raised during the review process.

We look forward to receiving your revised manuscript.

Kind regards,

Michelle Tye, Ph.D.

Academic Editor

PLOS ONE

Additional Editor Comments (if provided):

Congratulations to the authors on an important and well written paper. In addition to the reviewer's comments, just as a very minor note, it's usually good practice to report the kappa or some inter-rater agreement metrics for the title/abstract stage of screening and for the full text review phase - this speaks to the robustness of your systematic review methods. Also, what proportion of the full text papers did the 2nd/3rd reviews end up reviewing? Please include.

Journal Requirements: 

2. Please upload a new copy of Figure 4 as the detail is not clear. Please follow the link for more information: " ext-link-type="uri" xlink:type="simple">https://blogs.plos.org/plos/2019/06/looking-good-tips-for-creating-your-plos-figures-graphics/"
" ext-link-type="uri" xlink:type="simple">https://blogs.plos.org/plos/2019/06/looking-good-tips-for-creating-your-plos-figures-graphics/"

3. We note that Figure S2 in your submission contain copyrighted images. All PLOS content is published under the Creative Commons Attribution License (CC BY 4.0), which means that the manuscript, images, and Supporting Information files will be freely available online, and any third party is permitted to access, download, copy, distribute, and use these materials in any way, even commercially, with proper attribution. For more information, see our copyright guidelines: http://journals.plos.org/plosone/s/licenses-and-copyright.

a. You may seek permission from the original copyright holder of Figure S2 to publish the content specifically under the CC BY 4.0 license. 

Reviewers' comments:

Reviewer's Responses to Questions

**Comments to the Author**

1. Is the manuscript technically sound, and do the data support the conclusions?

Reviewer #1: Yes

Reviewer #2: Yes

2. Has the statistical analysis been performed appropriately and rigorously? 

Reviewer #1: N/A

Reviewer #2: N/A

3. Have the authors made all data underlying the findings in their manuscript fully available?

Reviewer #1: Yes

Reviewer #2: Yes

4. Is the manuscript presented in an intelligible fashion and written in standard English?

Reviewer #1: Yes

Reviewer #2: Yes

5. Review Comments to the Author

Reviewer #1: Thank you for the opportunity to review this article. This systematic review of literature on transgender and non-binary youth accessing gender affirming care synthesises a complex body of work effectively and generates some insightful overarching themes in this work. For the most part, I believe this article to be well written and makes a useful contribution to the field. However, there are a few areas where I feel some aspects of the work could be clarified.

Abstract:

- What is a 'third order construct'? Perhaps this is discipline specific technical knowledge you explain later? Possibly need to clarify?

- I would like to see a stronger statement in the abstract, but also somewhere in the paper itself, as to what this review adds to the existing body of work, beyond summarising what has already been written on the topic. I believe that it does offer important insights, but I would like to see these articulated more strongly.

- Following on from your systematic review, what are the future directions/recommendations for research and practice? Again, this could also be more strongly stated in the conclusion of the paper as well.

Structure:

- At first, I wasn't entirely sure about the start of the paper, and wanted to see a clearer statement of argument and aims before you got into definitions, but I do see how explaining terminology is important from the start. However, I was wondering if you could condense some of the first section on language and definitions into a shorter 'glossary of terms' to get straight to the point of the paper. A brief mention of the purpose/aims/argument of the paper at the beginning would frame the paper more strongly.

- Again, while relevant, I suggest streamlining the discussion of the prevalence of trans and gender diverse people if possible.

- I thought the articulation of the aims of the project on p. 7 could be expanded upon. What do you aim to achieve by synthesising the literature?

Method/Design:

- There is lots of information in the methods and design section that I did not entirely understand. While this may have been due to disciplinary differences, some of the technicalities could be more clearly explained. For example, what exactly is a 'meta-ethnography'? Why was this approach chosen? What was the rationale for the databases you chose to use? When you state that "search terms were informed by PEO's" what does that actually mean? What are MeSH terms and Boolean operators? How are these assisting in the process? Why did you choose to focus on qualitative studies only?

Findings:

- The synthesis of common themes across the literature was interesting and sound. However, I thought the way participant quotes from the studies were integrated could be clarified and more consistent. There are numerous points where quotes are included, but not closely engaged with. It is not always clear where these quotes come from (e.g. participant quotes on p. 27).

Analysis:

- While I found the discussion of the behavioural ecological model compelling, it comes a little late in the piece for me. Could this have been flagged/foreshadowed earlier in the paper?

Thank you again for the opportunity to engage with this work. I hope some of my points will be useful for developing the paper further.

Reviewer #2: Thank you for the opportunity to review this submission. This is an important piece of work that summarizes the important considerations in the literature as they relate dot trans and non-binary youth access to care. I have several suggestions on how to strengthen the manuscript and will outline these by section. I would like to commend the authors on the comprehensive nature of this analysis as it represents quite a significant undertaking.

1. (Language and Definitions, paragraph 2) "Transgender individuals often identify in a binary manner". This is a bit of a nebulous area, but the terms can often overlap, or be exclusive, depending on the individual (e.g. some trans people identify as non-binary, some non-binary people do not identify as trans). It might be more appropriate to state that they "may" identify in a binary manner (many more recent quantitative surveys are seeing significantly large groups of people who reject the binary.

2. (Language and Definitions, paragraph 4) Define the term "social transition" for the reader.

3. (Introduction, paragraph 6) How exactly is "age" identified as a barrier in reference 62 (i.e. what is it about age that acts as a barrier?).

4. (Introduction, paragraph 7) The term "healthcare individuals" seems like an odd choice of wording, I would suggest simply using "them" to refer back to providers.

5. (Introduction, paragraph 7) Use quotation marks around the term "gatekeeping".

6. (Methods, Search Strategy, paragraph 1) Define what "PEO's" means.

7. (Methods, Quality Appraisal, paragraph 1) Edit last sentence to read "Mixed methods studies were assessed using the qualitative CASP checklist as only qualitative data were used in the synthesis."

8. (Methods, Data Extraction and Data Synthesis, paragraph 3) What data were uploaded to the NVivo, specifically? Was it participant quotes included in the publications reviewed?

9. (Results, Search results, paragraph 1) "Sourced from other sources" reads awkwardly.

10. (Results, Table 1) There seems to be differences in how each study is summarized. I would suggest changing the "Country" heading to "Country and/or region" as some studies were done in very localized environments. Additionally, the Methodology/Study Design section includes information about analytic strategies in some publications, but excludes this in others. While some papers may not have delved deeply into this particular aspect of their methodology, I would suggest including this, as it informs how the studies interpreted results (among other things).

11. (Results, Gender identity and interpretation) Repetition in the statement "Notably, Carlile (2019) noted that..."

12. (Results, Findings, overall) While some of the quotes included names, identities, and ages in brackets, it might be beneficial to also include perhaps a study identifier. For example, you could include the last name of the first author at the end of the brackets (and include it solely in brackets for quotes that had no identifiers). This would allow the reader to see more easily where quotes were drawn from, to identify whether there was an over-reliance on some studies for results than others.

13. (Results, 1.2.2. Geographical threats) The latter half of this paragraph reads more as something that would be included in the Discussion. The sentence that begins with "Ideally,..." has no supporting reference and reads more as a statement for the Discussion, for example.

14. (Life on argument synthesis: The Rainbow Brick Road) This model seems very interesting, and summarizes the synthesis of findings well. The visual itself, though, seems like it could be further developed. An explanation of why a wheel is used instead of a pathway, for example, that would be more reflective of a road. Additionally, the placement of the personal, environmental, etc., components aren't fully explained. The model itself, currently, seems out of place, and is not well-explained in the middle of the manuscript.

15. (Discussion, paragraph 1) How was it arrived that there was a "sufficient dataset for synthesis"?

16. (Discussion, adapted Ryvickers' behavioural ecological model) In the text, there are explanations for Predisposing and Need factors, but a lack of explanation about enabling factors. They are alluded to in the sections about Social Environment and Neighbourhood (among others), but not as clearly laid out. Further, Ryvickers' model, following from Andersen, defines Needs factors as those that "include both the patient’s perception and provider’s evaluation of healthcare needs, including health status and diagnoses." The latter half of this definition is not touched upon in the manuscript and should be further elucidated and included in the model the authors put forward. Further, in the "Health outcomes" part of the model, there is no mention of the health outcomes as outlined in Ryvickers' model (e.g. changes in chronic illnesses, symptoms, etc.). With some reference to external sources, the authors' might be able to comment on improvements seen in patients, not just on access to care factors (which are also things that are included under Health Behaviours and not Outcomes in Ryvickers' model).

17. (Overall comments) Please copy edit the manuscript as there were several instances where:

-plurals were used incorrectly

-contractions were used outside of quotes

-acronyms were not defined at first use

Generally, I really like the in-depth approach taken to this article, but would like to see some further development in the theoretical models put forward.

Great work so far, congratulations to the authors!

6. PLOS authors have the option to publish the peer review history of their article (what does this mean?). If published, this will include your full peer review and any attached files.

Reviewer #1: No

Reviewer #2: No

---

## [Author Response · Author response to Decision Letter 0]

10 Aug 2021

Comment from editor: Your article cannot proceed until you upload a copy of the completed PRISMA checklist as Supporting Information.

Response: PRISMA Checklist is attached as S6_File. 

See cover letter for responses formatted in table format. Content below.

Dear Editor and Reviewers, 

On behalf of myself and the co-authors of this paper, I would like to thank you all for your thoughtful comments and expertise in reviewing this paper. 

I feel that each suggestion improved the overall manuscript. See below my responses to each query in a table format and a tracked changes and clean version of the document has been uploaded too. 

Should you require anything else, then please do not hesitate to reach out. 

Best, 

Sean Kearns 

Editor comments Responses

Congratulations to the authors on an important and well written paper. In addition to the reviewer's comments, just as a very minor note, it's usually good practice to report the kappa or some inter-rater agreement metrics for the title/abstract stage of screening and for the full text review phase - this speaks to the robustness of your systematic review methods. Also, what proportion of the full text papers did the 2nd/3rd reviews end up reviewing? Please include.

• Included Kappa calculations for inter-rater agreement for title/abstract screening and full text agreement. Thank you for this suggestion. 

• Updated the proportion of full text reviews. 2nd reviewer reviewed in detail 100% with first author and third review read all and was available for consultation and opinion. Consensus easily reached and was initially strong as per kappa.

Please ensure that your manuscript meets PLOS ONE's style requirements, including those for file naming. The PLOS ONE style templates can be found at …

• Manuscript reviewed in line with PLOS ONE’s style requirements. If there are any discrepancies, then please let us know and happy to edit.

Please upload a new copy of Figure 4 as the detail is not clear. 

• Uploaded a clearer option, let me know if needs any further sharpening

We note that Figure S2 in your submission contain copyrighted images.

• Removed from manuscript. Not vital to the submission. All subsequent numerical changes for supplementary documents made. 

Reviewer 1 comments responses 

Abstract:

- What is a 'third order construct'? Perhaps this is discipline specific technical knowledge you explain later? Possibly need to clarify?

- I would like to see a stronger statement in the abstract, but also somewhere in the paper itself, as to what this review adds to the existing body of work, beyond summarising what has already been written on the topic. I believe that it does offer important insights, but I would like to see these articulated more strongly.

- Following on from your systematic review, what are the future directions/recommendations for research and practice? Again, this could also be more strongly stated in the conclusion of the paper as well.

• A third order construct would be a commonly used term for meta-ethnographies. I added additional definitions to the synthesis section of the paper to explain the terms in more detail.

• In short, first order constructs are the direct quotes from the papers reviewed. Second order constructs are the papers authors’ views and interpretations from their analysis and third order constructs were the views and interpretations of the synthesis team expressed as key concepts.

• Third order constructs should be telling a stronger story than a second order or first on its own. Tracking of first to third is also present as a supplementary document for one of the themes to serve as an example of the process. As this is qualitative in nature, there is no mechanical checklist of how this synthesis is derived, it is interpretive but still should be trackable. 

• They then feed into a line of argument, which is our model that we created which would be common from meta ethnographies to produce a new conceptual model or theory or way of understanding. 

• Thank you for the comments on a stronger impact on language. I added more into the abstract and in the main piece and strengthened the conclusion. See also implications for practice and further research section. 

Structure:

- At first, I wasn't entirely sure about the start of the paper, and wanted to see a clearer statement of argument and aims before you got into definitions, but I do see how explaining terminology is important from the start. 

However, I was wondering if you could condense some of the first section on language and definitions into a shorter 'glossary of terms' to get straight to the point of the paper. 

A brief mention of the purpose/aims/argument of the paper at the beginning would frame the paper more strongly.

- Again, while relevant, I suggest streamlining the discussion of the prevalence of trans and gender diverse people if possible.

- I thought the articulation of the aims of the project on p. 7 could be expanded upon. What do you aim to achieve by synthesising the literature?

- There is lots of information in the methods and design section that I did not entirely understand. While this may have been due to disciplinary differences, some of the technicalities could be more clearly explained. For example, what exactly is a 'meta-ethnography'? 

Why was this approach chosen?

What was the rationale for the databases you chose to use?

When you state that "search terms were informed by PEO's" what does that actually mean? 

What are MeSH terms and Boolean operators? 

How are these assisting in the process? 

Why did you choose to focus on qualitative studies only?

• Thank you for the feedback on this section. On hearing this, I summarised this section into a glossary table which I hope allows for an overview for readers who need it and allow for a simple transition to the introduction for those who do not need it. I condensed it a little, but as it is a thorough systematic review, I wanted to keep depth to it too.

• Restructured the introduction so that the initial paragraph has a statement on the purpose of the paper, this is tangentially linked to the glossary of terms.

• Deleted sections on prevalence and streamlined to be more concise. 

• I expanded more on the aims into three pieces, linking to models that come into the paper at a later point.

• Updated- added more info on meta-ethnography.

• Rationale given.

• Rationale given.

• Overview of what PEO’s are and how they informed the search are updated.

• MeSH and Boolean terms are well described in a supplementary file so I didn’t repeat here. Added a sentence on how they assist the search strategy process. 

• Added a comment on why qualitatite studies only. 

Findings:

- The synthesis of common themes across the literature was interesting and sound. However, I thought the way participant quotes from the studies were integrated could be clarified and more consistent. There are numerous points where quotes are included, but not closely engaged with. It is not always clear where these quotes come from (e.g. participant quotes on p. 27).

• I updated where each quote came from by adding the name of the first author and year to show that the quotes came from a combination of the included studies. 

Analysis:

- While I found the discussion of the behavioural ecological model compelling, it comes a little late in the piece for me. Could this have been flagged/foreshadowed earlier in the paper?

• I added a little more of this to the abstract as a pre-lude and added it as one of the main aims for the synthesis piece so that it is foreshadowed more.

Reviewer 2 Comments Responses

1. (Language and Definitions, paragraph 2) "Transgender individuals often identify in a binary manner". This is a bit of a nebulous area, but the terms can often overlap, or be exclusive, depending on the individual (e.g. some trans people identify as non-binary, some non-binary people do not identify as trans). It might be more appropriate to state that they "may" identify in a binary manner (many more recent quantitative surveys are seeing significantly large groups of people who reject the binary.

• Very true. Thank you for these thoughtful suggestions. I have changed the term often to “may” and added in a small comment on increasing prevalence of non-binary identities and on the inherent overlap between the trans and non-binary umbrella terms in that they aren’t mutually exclusive.

• I also changed this section to a glossary format to allow a smoother transition into the introduction.

2. (Language and Definitions, paragraph 4) Define the term "social transition" for the reader.

• Updated to include a brief author interpretation of social transition 

3. (Introduction, paragraph 6) How exactly is "age" identified as a barrier in reference 62 (i.e. what is it about age that acts as a barrier?).

• “additional screening required for younger ages and parental consent often a requirement” – added to manuscript

4. (Introduction, paragraph 7) The term "healthcare individuals" seems like an odd choice of wording, I would suggest simply using "them" to refer back to providers.

• Changed to “them”

5. (Introduction, paragraph 7) Use quotation marks around the term "gatekeeping".

• Agreed and updated

6. (Methods, Search Strategy, paragraph 1) Define what "PEO's" means.

• Updated

7. (Methods, Quality Appraisal, paragraph 1) Edit last sentence to read "Mixed methods studies were assessed using the qualitative CASP checklist as only qualitative data were used in the synthesis."

• Updated

8. (Methods, Data Extraction and Data Synthesis, paragraph 3) What data were uploaded to the NVivo, specifically? Was it participant quotes included in the publications reviewed?

• The data uploaded consisted of all direct participant quotes from the publications reviewed. 

• A comment on this was added to the manuscript for clarity.

9. (Results, Search results, paragraph 1) "Sourced from other sources" reads awkwardly.

• Deleted as not needed – reads better now

10. (Results, Table 1) There seems to be differences in how each study is summarized. I would suggest changing the "Country" heading to "Country and/or region" as some studies were done in very localized environments. 

Additionally, the Methodology/Study Design section includes information about analytic strategies in some publications, but excludes this in others. While some papers may not have delved deeply into this particular aspect of their methodology, I would suggest including this, as it informs how the studies interpreted results (among other things).

• Updated.

11. (Results, Gender identity and interpretation) Repetition in the statement "Notably, Carlile (2019) noted that..."

• Thank you - updated

12. (Results, Findings, overall) While some of the quotes included names, identities, and ages in brackets, it might be beneficial to also include perhaps a study identifier. For example, you could include the last name of the first author at the end of the brackets (and include it solely in brackets for quotes that had no identifiers). This would allow the reader to see more easily where quotes were drawn from, to identify whether there was an over-reliance on some studies for results than others.

• Good suggestion, I have done this for all the direct quotes showcasing that there are direct quotes from a variety of the included studies.

13. (Results, 1.2.2. Geographical threats) The latter half of this paragraph reads more as something that would be included in the Discussion. The sentence that begins with "Ideally,..." has no supporting reference and reads more as a statement for the Discussion, for example.

• Moved later half to appropriate section in discussion.

14. (Life on argument synthesis: The Rainbow Brick Road) This model seems very interesting, and summarizes the synthesis of findings well. The visual itself, though, seems like it could be further developed. An explanation of why a wheel is used instead of a pathway, for example, that would be more reflective of a road. Additionally, the placement of the personal, environmental, etc., components aren't fully explained. The model itself, currently, seems out of place, and is not well-explained in the middle of the manuscript.

• The original concept for the diagram was more that the wheel represented non-linear roads in a spherical model. I re-drafted the diagram based on your suggestions. It is now five roads leading towards gender-affirming care with intersecting arrows representing the crossover and different paths. I feel this is more representative of the model overall, so thank you for this note. 

• The line of argument/new conceptual model would traditionally always come after the results, so I am hesitant to move it. I have foreshadowed it more in the abstract so as to introduce the idea earlier. I hope that this suffices. 

15. (Discussion, paragraph 1) How was it arrived that there was a "sufficient dataset for synthesis"?

• Removed the phrase “sufficient data set”.

16. (Discussion, adapted Ryvickers' behavioural ecological model) In the text, there are explanations for Predisposing and Need factors, but a lack of explanation about enabling factors. They are alluded to in the sections about Social Environment and Neighbourhood (among others), but not as clearly laid out. Further, Ryvickers' model, following from Andersen, defines Needs factors as those that "include both the patient’s perception and provider’s evaluation of healthcare needs, including health status and diagnoses." The latter half of this definition is not touched upon in the manuscript and should be further elucidated and included in the model the authors put forward. Further, in the "Health outcomes" part of the model, there is no mention of the health outcomes as outlined in Ryvickers' model (e.g. changes in chronic illnesses, symptoms, etc.). With some reference to external sources, the authors' might be able to comment on improvements seen in patients, not just on access to care factors (which are also things that are included under Health Behaviours and not Outcomes in Ryvickers' model).

• Updated in detail. See discussion. Re-structured significantly and added additional section on outcomes and behaviours.

17. (Overall comments) Please copy edit the manuscript as there were several instances where:

-plurals were used incorrectly

-contractions were used outside of quotes

-acronyms were not defined at first use

• Contractions reviewed.

• Acronyms reviewed.

• Plurals reviewed.

---

## [Editor Report · Decision Letter 1]

26 Aug 2021

Experiences of transgender and non-binary youth accessing gender-affirming care: a systematic review and meta-ethnography.

PONE-D-20-38613R1

Dear Dr. Kearns,

We’re pleased to inform you that your manuscript has been judged scientifically suitable for publication and will be formally accepted for publication once it meets all outstanding technical requirements.

Kind regards,

Michelle Torok, Ph.D.

Academic Editor

PLOS ONE
---

## [Editor Report · Acceptance letter]

31 Aug 2021

PONE-D-20-38613R1 

Experiences of transgender and non-binary youth accessing gender-affirming care: a systematic review and meta-ethnography. 

Dear Dr. Kearns:

I'm pleased to inform you that your manuscript has been deemed suitable for publication in PLOS ONE. Congratulations! Your manuscript is now with our production department. 

Kind regards, 

on behalf of

Dr. Michelle Torok 

Academic Editor

PLOS ONE